# Seasonal Variations and Sexual Differences in Home Range Sizes and Activity Patterns of Endangered Long-Tailed Gorals in South Korea

**DOI:** 10.3390/ani15010027

**Published:** 2024-12-26

**Authors:** Sangjin Lim, Maniram Banjade, Jaeyong Ahn, Dongju Song, Jangick Son, Yungchul Park

**Affiliations:** 1Institute of Forest Science, Kangwon National University, Chuncheon 24341, Republic of Korea; sangjin@kangwon.ac.kr; 2College of Forest and Environmental Sciences, Kangwon National University, Chuncheon 24341, Republic of Korea; mani88zoo@gmail.com; 3Yanggu Goral Restoration Center, Yanggu 24506, Republic of Korea; ajy12@hanmail.net; 4Korea National Park Service, Wonju 26466, Republic of Korea; sdjoo317@gmail.com (D.S.); lutra4015@naver.net (J.S.)

**Keywords:** conservation, crepuscular, endangered, GPS collar, long-tailed goral, management

## Abstract

This study aims to investigate the home ranges and activity patterns of the endangered long-tailed goral (*Naemorhedus caudatus*), an ungulate threatened by habitat loss, fragmentation, and human disturbances in South Korea. From 2014 to 2016, data from nine GPS-collared individuals (four males and five females) released across three regions (Yanggu, Seoraksan National Park, and Uljin) were analyzed. The study found that males had larger home ranges than females; the home range size also differed by season, with the smallest occurring in winter. Long-tailed gorals were mainly active during crepuscular and daylight hours, with no significant differences in activity between the sexes. To the best of our knowledge, this is the first detailed analysis of the home ranges and activity patterns of long-tailed gorals in these regions, providing crucial information for guiding conservation strategies to protect this endangered species.

## 1. Introduction

The long-tailed goral (*Naemorhedus caudatus*) (hereafter referred to as goral) is an ungulate that inhabits the mountains of northern and eastern Asia, including Russia, China, parts of the Himalayas, and the Korean Peninsula [1]. Owing to a rapid decline in population, this species is recognized as vulnerable by the International Union for Conservation of Nature [2] and as endangered under the Convention on International Trade in Endangered Species [3]. The principal reasons underlying the decline of goral populations are poaching and habitat destruction caused by rapid economic development, increasing urbanization, and expanding transportation networks [2,4,5]. Given the declining population, it is crucial to understand the fundamental aspects of their behavioral ecology, such as home range characteristics and activity patterns, to implement targeted protection measures.

In South Korea, gorals were once abundant in the inland areas of Gangwon and the eastern mountainous areas until the 1960s. However, goral populations have decreased significantly because of habitat destruction, fragmentation, and high mortality from heavy snowfall since the 1960s [6]. Currently, stable populations of at least 100 individuals exist in four key areas—the demilitarized zone, Yanggu/Inje, Seoraksan, and Uljin/Samcheok–and only small populations are scattered in other areas [7,8]. Recognizing the urgent need for conservation, the goral was designated as a natural monument and classified as a Class I endangered species in South Korea [4].

Understanding home ranges and activity patterns is crucial when a taxon is threatened and relies on targeted conservation measures. Observing these patterns provides information on how they utilize their space, interact with competitors, regulate their temperature, and adapt to their surrounding environments [8]. However, such an ecological study of gorals through direct observation is difficult because of their low density and elusive nature. The introduction of the GPS collar has revolutionized research [9] because of its low operating cost and high efficiency [10]

Gorals are good examples of medium-sized, cryptic, and uncommon ungulate species. Similar to most mountain ungulates worldwide, the ecology of gorals has been less studied in their distribution ranges outside of South Korea [11]. However, ecological studies in South Korea are restricted to habitat suitability, genetic diversity, population composition, sex identification, and, to a certain extent, home range [12,13,14,15,16]. Therefore, this study aims to assess the home ranges and activity patterns of gorals by sex and season using GPS collars. We hypothesize that the home range characteristics and activity patterns differ significantly between male and female gorals across seasonal contexts, revealing key insights into their spatial ecology and behavioral dynamics.

## 2. Materials and Methods

### 2.1. Study Area

This study was conducted in three distinct mountainous regions along the east coast of the South Korean Peninsula (the Yanggu, Seoraksan National Park, and Uljin areas; Figure 1). These regions represent three of the four major habitats of gorals in South Korea. The temperatures of these regions typically range from −10 °C to 25 °C, with cold winters often falling below freezing and warm, humid summers. The typical annual precipitation ranges from 1000 to 2000 mm, contributing to the overall forest structure and influencing the availability of food resources. The terrain across the entire area is predominantly rugged, steep, and covered with broadleaf deciduous trees and coniferous species [14]. All study areas are home to gorals, roe deer (*Capreolus pygargus*), water deer (*Hydropotes inermis*), wild boars (*Sus scrofa*), raccoon dogs (*Nyctereutes procyonoides*), and Asian badgers (*Meles leucurus*). No predatory animals were present in the study area.

### 2.2. GPS Monitoring

Our study was conducted on 11 gorals (all adults; rescued individuals raised at a restoration center) that were released between 2014 and 2016 (Table 1). Each goral was fitted with a GPS collar (Global Star Track S, Lotek, Newmarket, ON, Canada, Wt. 420 gm), which is approximately 2% of the average body weight, attached to its neck. We employed IC-R20 receivers (ICom Inc., Osaka, Japan) and 3-element Yagi-antennas (ATS Inc., Isanti, MN, USA) to monitor goral movement when GPS data were unavailable. To enhance the accuracy and reliability of the goral location information, we used standard triangulation techniques [17], and their Universal Transverse Mercator coordinates were estimated using the AniMove plugin in QGIS (version 0.2.3). Eight of the collared individuals were released at two locations in Yanggu, two in Seoraksan, and one in Uljin. The released individuals included six males and five females. The GPS collar fitted to the goral automatically recorded location data every four hours. These collars were programmed to transmit GPS coordinates via satellite, allowing us to remotely collect data without manual fieldwork. To minimize behavioral biases related to acclimation, the data analysis began one month after their release. Animal handling was performed according to the guidelines of the American Society of Mammologists [18].

### 2.3. Data Analysis

To estimate the home ranges, we only included locations that were statistically independent, as determined by the Schoener index [19], using a threshold of 0.5. This threshold is commonly used in spatial ecology studies to minimize autocorrelation in movement data [20,21]. Although the small sample size and distribution by gender may limit the representativeness of the data for the broader population, we took steps to minimize potential biases. Before combining the GPS data from the three regions, we conducted statistical analyses to check the differences in the environmental conditions (e.g., altitude, vegetation type, and proximity to human settlements) and demographic factors (e.g., age and sex) among the regions. These analyses revealed no significant differences, allowing us to combine the data for a more comprehensive analysis. We calculated the home range of each individual using both the 95% minimum convex polygon (MCP) and the 50% MCP (core area) employing the Home Range Tool extension in ArcGIS 10.2 (ESR, Redlands, CA, USA). Welch’s *t*-test [22] was used to compare the average and seasonal home range sizes based on sex. Seasonal variations in the average home range sizes were analyzed via analysis of variance. To assess differences in the likelihood of a goral being active based on the time of day, sex, and season, we utilized the Wald chi-square test [23]. Statistical significance was set at *p* < 0.05. We used the criteria from Bennie et al. [24] to classify each individual’s activity as diurnal (active during the day), nocturnal (active at night), crepuscular (active during twilight, which include dawn and dusk), or cathemeral (exhibiting no specific time preferences). We defined the seasons as spring (March–May), summer (June–August), autumn (September–November), and winter (December–February). To determine the activity patterns of the studied species, we analyzed GPS collar location data recorded 5–6 times per day. We recorded individual activity based on data from the motion sensor accelerometer integrated into the GPS collars, coding the activity as a binary variable (1 for active and 0 for inactive) in relation to the time of day, sex, and season. Activity patterns were analyzed using the non-parametric Kernel density approach [25] to estimate the activity patterns of the species across seasons.

## 3. Results

### 3.1. Home Range

We recorded 6286 independent locations from 11 individual gorals (6 males and 5 females) fitted with GPS collars. The coordinate points from two individuals (IDs #G4 and #G5) were excluded from the analysis because they were collected for less than one year. However, data from nine individuals (four males and five females) were collected sequentially throughout the year-long survey period. Observation revealed that, after release, the animals initially exhibited stationary movement for a few days (almost 3–4 days) and then exploratory behavior. According to the 95% MCP analysis, the average home range was 0.64 ± 0.33 km^2^, while the 50% MCP core home range was 0.15 ± 0.05 km^2^. The size of the home range (MCP 95%) was largest for individual #G7 (1.38 km^2^) and smallest for individual #G9 (0.31 km^2^). Regarding the core habitats (MCP 50%), individual #G3 had the largest area, at 0.25 km^2^, whereas the smallest area of 0.10 km^2^ was observed for individual #G1. In general, gorals tend to have one or more 50% MCP home range centers, depending on the species and individual behaviors. Their spatial use patterns can vary, but studies typically report an average home range size of 0.15 ± 0.05 km^2^, indicating a relatively small and concentrated area of activity.

In winter, the seasonal home ranges of males and females did not differ significantly (*t* = 0.38, *p* = 0.072), and the winter home ranges, calculated using the 95% MCP, were very similar for both sexes. However, there was a significant difference in the core area (50% MCP) during winter, with males exhibiting larger core areas compared to female (*t* = 1.23, *p* = 0.042). Considering the entire study period, males exhibited a substantially broader home range than females (*t*-test, *t* = 2.10, *p* = 0.04; Table 2). The size of the home range for males was 1.3 times larger than that for females when measured using the 95% MCP, and 0.75 times larger when measured using the 50% MCP. Across all seasons, the goral home range was the smallest during winter (Table 2). The typical home ranges (MCP) for two gorals (IDs #G2 and #G3) are presented in Figure 2. An overview of the seasonal home ranges and test statistics for each sex is listed in Table 2.

### 3.2. Activity Pattern

The kernel density estimates of the goral activity indicate crepuscular and/or diurnal activity throughout the study period (Figure 3). The circadian period exhibited significant variation in goral activity, with the highest bimodal activity peaks observed at approximately 05:00–08:00 and 17:00–20:00. Conversely, the lowest activity was recorded during the night (χ2 = 12.14, *p* = 0.003). The activity patterns of the sexes did not differ significantly (χ2 = 2.17, *p* = 0.08); however, males were more active than females in nearly every season (Figure 3). There was no significant variation in goral activity throughout spring, summer, and autumn; however, activity in winter was significantly different from that in the other seasons (Table 3). During winter, daytime activity was significantly higher than nighttime activity.

## 4. Discussion

Understanding the home range sizes and activity patterns of vulnerable species is crucial for effective management, especially in developed countries such as South Korea, where wildlife is increasingly affected by natural calamities and human activities. The goral, a species adapted to rugged mountainous terrains, faces challenges associated with these pressures, making detailed behavioral data essential for effective management. Our analysis reveals that the goral has a home range size averaging 0.39 ± 0.26 km^2^, with distinct activity peaks at dawn and dusk. These patterns are influenced by the complex mountainous landscapes and varying habitat conditions. Mountainous terrain with steep slopes, rocks, and variable elevations offers gorals both forage availability and protection from threats. Steep slopes and rugged areas provide shelter and thermal regulation, while different vegetation zones at different elevations provide seasonal food resources, requiring movement across a range of elevations [26]. These habitat features directly influence the home range size, as gorals must balance access to resources with the need for safety. Similarly, the dawn and dusk activity peaks align with times when foraging can occur at cooler temperatures. So, gorals’ spatial and temporal behaviors are adaptations to the challenges and opportunities of their mountain habitat.

This study utilized GPS collars to collect behavioral data on the goral’s home range to facilitate the conservation of this endangered species. In this study, the average 95% MCP home range was 0.64 ± 0.33 km^2^, while the 50% MCP was 0.15 ± 0.05 km^2^. The range observed in the present study was smaller to those observed for gorals in other parts of South Korea [13,27]. Several factors may explain this difference in home range sizes. In previous studies, the home ranges were estimated using a small number of individuals, which could have led to biased results and overestimation [28]. Furthermore, altitude and geographical location are important variables. Cho et al. [27] reported that gorals in Woraksan National Park were found in areas with altitudes ranging from 800 to 1200 m, characterized by steeper, more rugged terrain that supports diverse vegetation and likely provides abundant resources and shelter. In comparison, in our study, the goral populations were found in areas with altitudes ranging from 600 to 800 m. These areas have a moderately steep terrain with varying levels of vegetation cover. In addition, the individuals in this study were rehabilitated prior to their release. Rehabilitated animals have smaller home ranges compared to wild-born animals. This may be due their limited exposure to natural environments, leading to less exploratory behavior and use of habitat [29]. Rehabilitation may also impact their ability to find and use resources over a larger area, especially if they were habituated to human care during captivity [30]. Studies on other rehabilitated mammalian species have reported restricted movement and smaller home ranges after release due to the combination of behavioral adaptations and environmental unfamiliarity [31,32]. Hence, caution is needed when generalizing these results to other goral populations.

Our study identified sex-based differences in home ranges, with male gorals having larger home ranges than female gorals. Similar sexual variations in home ranges have been noted for gorals in South Korea [33]. We believe that the extent of the variations in the male and female movement rates [34] may sufficiently explain the sex differences in the annual home range sizes. Breeding and birth cycles may have contributed to the decline in female home ranges observed in our study. Cho et al. [33] found that mature females exhibited a decrease in home range size that coincided with the birth of their young. For instance, in red deer (*Cervus elaphus*) and white-tailed deer (*Odocoileus virginianus*), males have larger home ranges during the breeding season as they search for estrous females, while females reduce their movements during calving or fawning seasons to prioritize offspring care and safety [35,36]. Similarly, in bighorn sheep (*Ovis canadensis*), males expand their ranges during the rut to find multiple mates, while females focus on secure, resource-rich areas for raising their young [37].

During winter, herbivores typically expand their home range size in response to resource scarcity [38]. However, contrary to our expectations, the gorals’ home ranges were considerably smaller in winter than in the other seasons (Table 2). This pattern is consistent with other mountain ungulates, such as mountain goats (*Oreamnos americanus*) and Rocky Mountain bighorn sheep (*Ovis canadensis canadensis*), both of which reduce their home range sizes in winter to conserve energy and cope with the challenges of navigating snow-covered terrain and limited food resources [39,40]. In particular, for mountain ungulates, physiological and behavioral modifications are necessary for winter survival [41], because movement costs can be exceedingly high owing to the low temperatures in mountainous areas. The small home ranges may also be attributed to the decreased availability of food resources during winter. In response to these challenges, it is necessary to conserve energy by reducing locomotion during periods of resource scarcity and compensating for decreased food intake throughout the winter [42].

In addition, basic habitat conditions and climatic factors play an important role in limiting movement during winter [43]. Snow depth and icy terrain may further restrict movement, forcing goral to stay in sheltered or less exposed areas that provide better protection [44]. Furthermore, habitat characteristics like slope, aspect, and vegetation cover may alter the availability of feeding locations and movement pathway contributing to the reduced home range during winter. Future studies should focus on how these environmental variables influence goral movement and home range size throughout the winter season.

The gorals exhibited predominantly crepuscular and diurnal activity patterns, exhibiting bimodal activity peaks around sunrise and sunset. Our results are similar to those for other goral populations in South Korea [45] and China [5,46]. Such circadian activity patterns, with prominent peaks during crepuscular hours and daytime, have been observed in most wild ungulates [5,47,48]. Notably, our study’s species did not exhibit significant sexual differences in activity patterns, which aligns with the findings in other ungulates, such as red deer [49], roe deer [50], and red muntjac [51]. Sexual differences in activity patterns among ungulates are often associated with differences in body size. Typically, males and females conduct different activities owing to allometric differences in body size, resulting in variations in dietary requirements and energy expenditure [52]. However, these differences were not evident in the gorals. This lack of sexual dimorphism in activity patterns could be attributed to the relatively minor differences in body size between males and females in this species [15,53].

In the present study, no significant variations in goral activity patterns were observed during spring, summer, or autumn. However, the winter activity patterns differed significantly from those of the other three seasons (Table 2). The overall activity patterns revealed that, in winter, both male and female gorals were more active during the day, most likely in response to extreme daily temperature fluctuations. This behavior is presumed to help reduce energy expenditure and ensure sufficient food intake in low environmental temperatures [54,55]. During winter, the surrounding environmental temperature is low, and deer attempt to maintain their daily cycle by moving during the warmest part of the day. A previous study showed that ungulates exhibit nocturnal hypometabolism during winter to preserve energy [56,57], indicating that this phenomenon may have resulted in reduced activity levels of the goral during the coldest parts of the day (early morning) and night.

Our investigation of the home ranges and activity patterns of nine gorals (four males and five females) in three specific regions of South Korea (Yanggu, Seoraksan, and Uljin) provides unique insights into the species behavior in diverse habitat. However, our study is limited by a sample size of nine individuals, which may be insufficient to reflect the diversity in behavior across the population. In addition, the data collection effort was limited to three specific regions, which may limit the generalizability of our findings to other areas of the gorals’ range. Moreover, as the individuals were rehabilitated, their behavior may differ from that of wild individuals. Therefore, further studies should include wild individuals, large sample sizes, and a broad range of habitat to better understand sexual differences in home ranges and activity patterns. Additionally, data collection efforts should focus on habitat suitability, mating strategies, and dietary preferences, all of which may influence changes in home ranges and activity patterns. Understanding these aspects is crucial for designing successful conservation strategies and ensuring the long-term survival of this species in its native habitat.

## 5. Conclusions

Our study offers valuable insight into the seasonal home ranges and activity patterns of gorals in South Korea, which align with the typical spatial and activity patterns observed in mountain ungulates. The results show that male gorals had larger home ranges than female gorals, with the smallest home ranges observed in winter. Activity patterns were generally crepuscular and diurnal, exhibiting significant seasonal variation, with notably reduced activity during winter. Despite the limitations, our study emphasizes the considerable difference between seasons and sexes. Although the individuals were rehabilitated, we extracted valuable information on the spatial and temporal behaviors of long-tailed gorals in South Korea. This study contributes greatly to our understanding of the ecological requirements of gorals and provides a foundation for effective conservation planning in South Korea and other regions facing comparable conservation issues.

## Figures and Tables

**Figure 1 animals-15-00027-f001:**
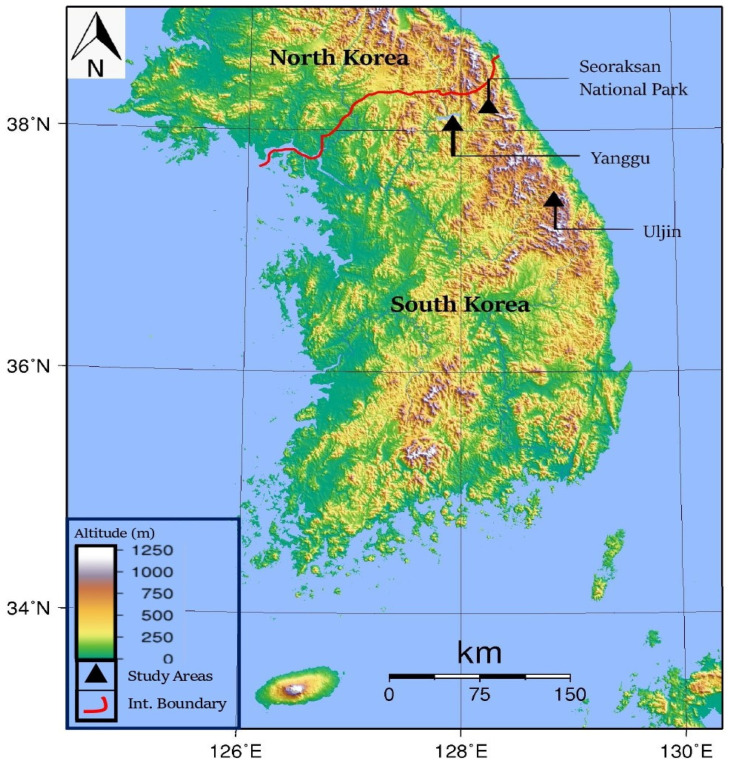
Elevation map of the Korean Peninsula including the geographical location of the three study sites.

**Figure 2 animals-15-00027-f002:**
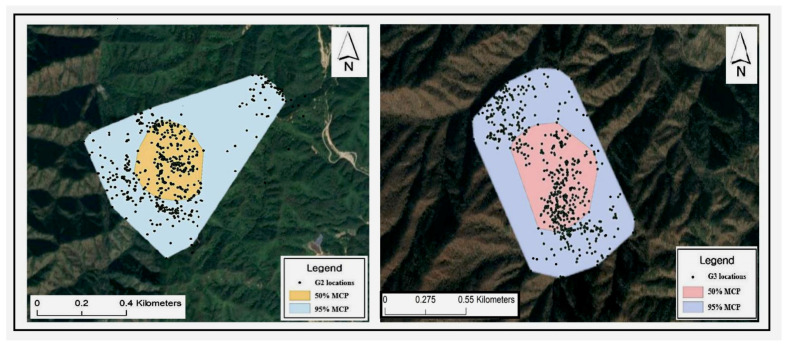
The 95% and 50% minimum convex polygon (MCP) home ranges of two long-tailed gorals (IDs #G2 and #G3) observed at the study site.

**Figure 3 animals-15-00027-f003:**
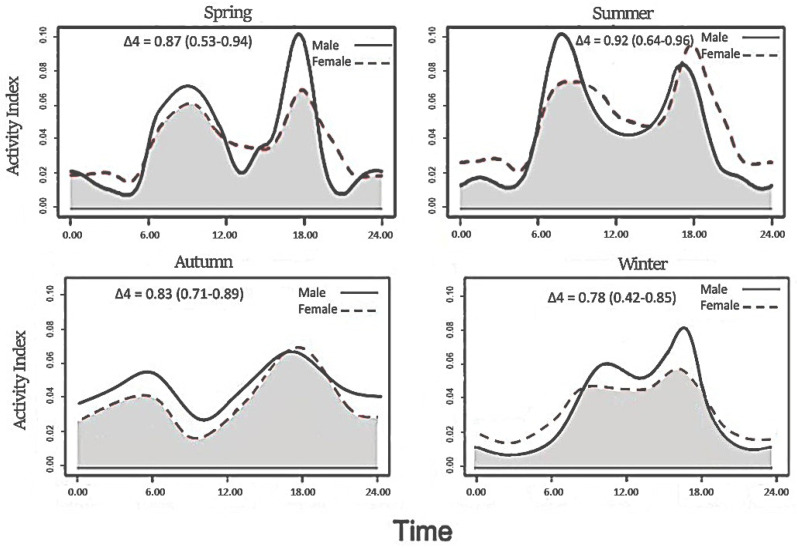
Seasonal activity patterns of male and female long-tailed gorals observed at the study sites. Solid and dashed lines indicate kernel density estimates for the male and female gorals, respectively. The grey shaded area in each plot represents the overlap coefficient. *X*-axis: time of day in hours; *Y*-axis: activity index.

**Table 1 animals-15-00027-t001:** Information on 11 gorals fitted with a GPS collar. The locations YG, SNP, and ULJ represent Yanggu, Seoraksan National Park, and Uljin, respectively. * Indicates individuals whose data were not included in further analysis.

Individual ID	Released Date (Location)	Sex	Weight (kg)	Age	Tracking Period	No. of Locations
G1	2014 May (YG)	F	32.4	3	2014 May~2015 July	374
G2	2014 May (YG)	M	39.7	4	2014 May~2015 July	572
G3	2014 June (ULJ)	F	36.2	2	2014 June~2015 Aug.	640
G4 *	2015 May (YG)	M	31.2	4	2015 June~2016 Jan.	214
G5 *	2015 May (YG)	M	37.8	3	2015 June~2016 Mar.	303
G6	2015 June (YG)	M	41.1	5	2015 June~2016 Aug.	544
G7	2015 June (YG)	M	35.3	3	2015 June~2016 Aug.	724
G8	2015 June (YG)	F	38.0	3	2015 June~2016 Aug.	496
G9	2015 June (YG)	F	34.1	4	2015 June~2016 Aug.	678
G10	2016 May (SNP)	F	33.5	4	2016 May~2018 Apr.	863
G11	2016 May (SNP)	M	37.6	3	2016 May~2018 Feb.	878

**Table 2 animals-15-00027-t002:** Seasonal home range for the gorals according to the 95% and 50% minimum convex polygons (MCPs) at the study sites. The comparison was conducted using the *t*-test. Data are expressed as the mean (±SE).

Season	No. Locations	Method	Male (*n* = 4)	Female (*n* = 5)	Sex Difference	
					*t*	*p*
Spring	1902	MCP_95_	0.61 ± 1.4	0.32 ± 0.4	2.17	0.014 *
		MCP_50_	0.15 ± 0.01	0.11 ± 0.01	1.42	0.021 *
Summer	2002	MCP_95_	0.32 ± 0.2	0.19 ± 0.1	1.69	0.013 *
		MCP_50_	0.20 ± 0.03	0.05 ±0.3	2.84	0.002 *
Autumn	1243	MCP_95_	0.15 ± 1.2	0.12 ± 0.01	2.78	0.001 *
		MCP_50_	0.07 ± 0.1	0.04 ± 0.1	1.94	0.018 *
Winter	1139	MCP_95_	0.22 ± 0.6	0.21 ± 0.06	0.38	0.072
		MCP_50_	0.11 ± 0.17	0.06 ± 0.7	1.23	0.042 *

* α, with a significant difference of < 0.05.

**Table 3 animals-15-00027-t003:** Seasonal patterns of the activity of the gorals. The values in the table represent trap locations (i.e., recorded positions of individual gorals fitted with a GPS collar) ± S.E.

Seasons	Trap Locations ± S.E						*p*-Value	
	Twilight		Daytime		Nighttime		M	F
	M	F	M	F	M	F		
Spring	26.2 ± 1.20	22.1 ± 10	21.3 ± 0.3	20.3 ± 0.43	18.5 ± 0.21	16.0 ± 0.60	0.15	0.17
Summer	27.2 ± 0.18	18.3 ± 0.27	22.3 ± 0.84	14.1 ± 0.16	20.8 ± 0.01	16.4 ± 0.11	0.42	0.12
Autumn	20.7 ± 0.16	23.5 ± 0.15	21.3 ± 0.17	21.5 ± 0.11	18.3 ± 11	19.9 ± 0.07	0.32	0.26
Winter	16.1 ± 0.31	14.5 ± 1.2	12.2 ± 0.27	8.9 ± 0.12	8.6 ± 0.21	6.3 ± 0.16	0.006 *****	0.03 *****
*p*-Value	0.005 *****	0.03 *****	0.16	0.21	0.13	0.21		

M = male; F = female; S.E = standard error. * α, with a significant difference of <0.05.

## Data Availability

The data that support the results of this study are accessible upon request from the corresponding author.

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
