# Peer review of "Seasonal Variations and Sexual Differences in Home Range Sizes and Activity Patterns of Endangered Long-Tailed Gorals in South Korea"

_animals, 2024, doi:10.3390/ani15010027_

Round 1
Reviewer 1 Report (Previous Reviewer 3)
Comments and Suggestions for Authors
I have reviewed the author's response to the comments and the changes made in the text, and I don't think there's anything more that needs to be changed, so I'd recommend accepting it.
Author Response
Thank you for your comments
Reviewer 2 Report (New Reviewer)
Comments and Suggestions for Authors
It should be noted that the authors undertook very interesting research into the functioning of the population of an endangered species. These studies show the functioning of the population in changing environmental conditions, as well as the shaping of animal behavior depending on the habitat and season, which is of great importance when taking protective measures. Although the results obtained in many aspects confirm the current knowledge in the field of ecology and behavior of this species, they are valuable in that they highlight the current state of knowledge in the field of behavior, which, in terms of problems related to the scale of threat to the species, is valuable in terms of undertaking rescue actions.
In my opinion, the results are interesting (even though nothing new was actually found) and are worth publishing. However, I have some comments that do not diminish the research and its results, but in my opinion the study should be slightly revised.
These notes are:
1. The introduction does not clearly specify the purpose of the work, only a description of what the results are intended to be used for. The research hypothesis must be clearly formulated.
2. In the case of methodology, it is worth paying attention to the fact that the research included 11 individuals in 3 locations, divided by gender, the sample in each of them was not representative and the results obtained, in my opinion, cannot fully reflect the actual situation, as they emphasize authors in the discussion. Moreover, as the authors say, these were animals raised in captivity, can their behavior be considered and compared with wild animals? In my opinion, this should be mentioned in the study, because the results of behavioral studies of animals raised by humans and wild animals may differ quite significantly in many aspects.
3. Results, why the tracking periods were ended so quickly, whether the batteries failed, whether this was the research plan. I can't find it?
4. In the summaries and methodology, the years 2014-216 are given, and in Table 1 in two cases the year is 2018?
5. The discussion is too long and, in many places, it is actually a repetition of the results (there are even references to tables, completely unnecessary). This chapter requires major revision. The authors should focus on their results, but against the background of others previously published for this species and this or a similar habitat. The last paragraph of the discussion is unnecessary.
6. I did not find information about the consent of the ethics committee, is such consent not necessary in South Korea?
7. Why did the authors decide to publish the results so late, since the research ended 8 (or 6) years ago?
I leave the decision whether to take my comments into account to the Academic Editor, and I recommend the manuscript for printing.
Author Response
To Editor
Animal Journal
Sub: Response to Reviewers
I am responding on behave of manuscript entitled “Seasonal variation and sexual differences in home range size and activity patterns of endangered long-tailed gorals in South Korea” animals-3352277. The manuscript was sent for major revision and we hereby respond the suggestion provided by reviewers as below:
Note: Changes are highlight with yellow color in the main text.
Reviewer #2:
Q 1. The introduction does not clearly specify the purpose of the work, only a description of what the results are intended to be used for. The research hypothesis must be clearly formulated.
Response: Thank you for your comments. We now clearly stated the study purpose and hypothesis of our study. Please refer to L. 80-82.
Q 2. In the case of methodology, it is worth paying attention to the fact that the research included 11 individuals in 3 locations, divided by gender, the sample in each of them was not representative and the results obtained, in my opinion, cannot fully reflect the actual situation, as they emphasize authors in the discussion. Moreover, as the authors say, these were animals raised in captivity, can their behavior be considered and compared with wild animals? In my opinion, this should be mentioned in the study, because the results of behavioral studies of animals raised by humans and wild animals may differ quite significantly in many aspects.
Response: Thank you for your comments. In response to your concern about the sample size and representativeness, we would like to clarify that before combining GPS data from the three regions, we conducted statistical analyses to assess potential differences in environmental conditions (e.g., altitude, vegetation type, proximity to human settlements) and demographic factors (e.g., age and sex) among the regions. The results of these analyses indicated that there were no significant differences. Therefore, we combined the data from the three regions for the analysis. We acknowledge the limitation, and we have address this in the discussion section. Please refer to L. 300-307.
Yes, we agree with your comments regarding the 2nd questions. We would like to clarify that the released individuals were not fully raised in captive since birth. Rather, they were rescued from the wild and were kept in a captive environment for a short period until they fully recovered. Once in good condition, they were immediately released back into the wild. We have included this in discussion section and as a limitation in our study. Please refer to L 227-237 and L. 314-315.
Q 3. Results, why the tracking periods were ended so quickly, whether the batteries failed, whether this was the research plan. I can't find it?
Response: Thank you for your comments. The tracking period ended so quickly because of GPS collar being drop off quickly due to some technical error.
Q 4. In the summaries and methodology, the years 2014-216 are given, and in Table 1 in two cases the year is 2018?
Response: Yes, in the summaries and methodologies we have stated that the individual was released between 2014-2016. However, we have chosen not to specify the tracking period, which was until 2018 in two cases.
Q 5. The discussion is too long and, in many places, it is actually a repetition of the results (there are even references to tables, completely unnecessary). This chapter requires major revision. The authors should focus on their results, but against the background of others previously published for this species and this or a similar habitat. The last paragraph of the discussion is unnecessary.
Response: We have refined the discussion section. We have compared our result and discussion with other species also. Please refer to discussion section. Sorry to say the last paragraphs contains limitation of the study, hence we feel that this paragraph is necessary.
Q 6. I did not find information about the consent of the ethics committee, is such consent not necessary in South Korea?
Response: Thank you for your concern. We have already forwarded the ethical committee consent letter to the journal editor.
- Why did the authors decide to publish the results so late, since the research ended 8 (or 6) years ago?
Response: Thank you for your concern. The delay in publishing the results was due some personal reasons.
Thank you
Reviewer 3 Report (New Reviewer)
Comments and Suggestions for Authors
Simple summary
long-tailed goral (Naemorhedus caudatus), an ungulate which is threatened by
I would only mention the number of analysed individuals, otherwise it is a little confusing
Abstract
I would mention IUCN classification VU too
Keywords: I miss conservation as a important keyword
Introduction
General distribution and decline of the goral is clear
Situation in South Korea is hard to read. In my opinion there is no need for a detailed description what ministry innated what program. A general description about conservation actions and needs in South Korea would be enough.
I would start with line 72-78 and continue with 63-69. There is no need to describe the application of collar in the introduction.
Therefore, this study aims to 78 assess the home ranges and activity patterns of gorals by sex and season using GPS collars. These findings are expected to provide valuable insights to governments and conservation agencies for developing effective management and conservation strategies – these results are important for everybody
Methods
Study area – temperature range is described very rough
There is no need to describe the occurrence of other species in such detail. black bears (Ursus thibetanus) occurs but there are no predators present?
Fig 1 is hard to read. Small map in the upper left corner includes two maps, which is not clear. Colours are misleading. Colouring wat in blue would make it easier to understand. The large maps show provinces? Is this important. The border between North and South Korea is hard to find. I would be more important to have a map showing the mountain range or at least elevations. The ecological situation would be more important than the political situation.
GPS monitoring
Would be interesting the know the weight of the animals and of the transmitters. The use of rehabilitated animals limits the study and should be mentioned at least in the abstract
Line 125 Fig. 1 does not refer to location data. Was to location interval 4 or 5 hours. If it is different related to animals mention it in Table 1 if it depends on season or situation describe it.
Define activity of goral estimated using GPS data
Line 135 define environmental parameters or skip it because not presented in the results
Results
Line 157 - Detailed information on the individual gorals fitted with 157 GPS collars is listed in Table 1. There is no need for this sentence. Already described in methods.
I miss a qualitative description of the spatial distribution. Do all gorals have a single home range centre (50% MCP)?
After following the animals for the first month, was their movement stable or did they become stationary directly after the release?
Table 2. Describe the test used to compare home range size in the description of the table.
Figure 3. Seasonal activity pattern of male and female long-tailed goral observed at study sites. Solid and dashed lines indicate Kernel density estimates for male and female goral respectively. (Already described in the figure). The grey shaded area in each plot represents the overlap coefficient. X-axis: time of day in hours, Y-axis: activity index. (already described in the figure)
How differs the information between Fig.3 and Table 3. In my opinion they show the same result.
Discussion
Line 205 - especially in developed countries such as South Korea, - it is general important
How is home range size and activity pattern influenced by the complex 210 mountainous landscapes and varying habitat conditions. This is a very general assumption without explanation
Line 220 in other parts of South Korea?
Line 230 – you did not present you analysis about the influence of environmental factors. In my opinion you cannot use this results in the discussion. You have to discuss the influence of using rehabilitated animals on the results of your study.
Is the sexual based difference in home range size know form other ungulate species, how is it related to social system, is it normal to other mammals with the same social system?
Line 256 the same as above. Is this known for other mountain ungulates too?
The discussion is very general and superficial. I miss more general information about mountain ungulate behaviour and ecology.
Conclusion
Line 303 does the study really provide a comprehensive understanding after you described the limitation Line 289-301
The conclusion is more a summary than a conclusion and again you never mentioned the use of rehabilitated animals.
My biggest concern is the use of rehabilitated animals only, which is only mentioned in the methods and the presentation of the results as comprehensive.
Author Response
To Editor
Animal Journal
Sub: Response to Reviewers
I am responding on behave of manuscript entitled “Seasonal variation and sexual differences in home range size and activity patterns of endangered long-tailed gorals in South Korea” animals-3352277. The manuscript was sent for major revision and we hereby respond the suggestion provided by reviewers as below:
Note: Changes are highlight with yellow color in the main text.
Reviewer #3:
Simple summary
Q 1. long-tailed goral (Naemorhedus caudatus), an ungulate which is threatened by
I would only mention the number of analysed individuals, otherwise it is a little confusing
Response: Thank you for your comments. We have revised the simple summary version so as to include only the analysed individuals. Please refer to L. 13-14.
Abstract
Q 2. I would mention IUCN classification VU too.
Response: Thank you for your comments. We have included the IUCN category also. Please refer to L. 23.
Keywords
Q 3. I miss conservation as a important keyword
Response: Based on your comments, we have included the important but missing keywords “conservation”. Please refer to keywords.
Introduction
Q 4. General distribution and decline of the goral is clear but Situation in South Korea is hard to read. In my opinion there is no need for a detailed description what ministry innated what program. A general description about conservation actions and needs in South Korea would be enough.
Response: Thank you for your concern. Based on your suggestion we have removed the initiation effort of ministry. Please refere to L.58
Q 5. I would start with line 72-78 and continue with 63-69. There is no need to describe the application of collar in the introduction. Therefore, this study aims to 78 assess the home ranges and activity patterns of gorals by sex and season using GPS collars. These findings are expected to provide valuable insights to governments and conservation agencies for developing effective management and conservation strategies – these results are important for everybody
Response: Based on your comments we have removed the application of GPS collar in the introduction section. Please refer to L. 68.
Methods
Study area
Q 6. temperature range is described very rough
Response: We have managed the temperature range of the study area. Please refer to L 86.
Q 7. There is no need to describe the occurrence of other species in such detail. black bears (Ursus thibetanus) occurs but there are no predators present?
Response: In this section we only mentioned the name of some other co-occurring species, which may impact their home range size and activity pattern. However, regarding black bears (Ursus thibetanus) sighting was much rare so we removed its name.
Q 8. Fig 1 is hard to read. Small map in the upper left corner includes two maps, which is not clear. Colours are misleading. Colouring wat in blue would make it easier to understand. The large maps show provinces? Is this important. The border between North and South Korea is hard to find. I would be more important to have a map showing the mountain range or at least elevations. The ecological situation would be more important than the political situation.
Response: Thank you for your comments. We have made changes to figure 1 and shows altitude ranges. Please refer to Fig. 1.
GPS monitoring
Q 9. Would be interesting the know the weight of the animals and of the transmitters.
Response: Thank you for your comments. We have mentioned the weight of individual animals in the Table 1 and transmitter weight as 420 gm, which is mentioned in the manuscript.
Q 10. The use of rehabilitated animals limits the study and should be mentioned at least in the abstract.
Response: Thank you for your suggestions. We have added the point rehabilitated in the abstract. Please refer to L.25.
Q 11. Line 125 Fig. 1 does not refer to location data. Was to location interval 4 or 5 hours. If it is different related to animals mention it in Table 1 if it depends on season or situation describe it.
Response: We apologize for the mistake. We have removed the denotation as Fig. 1, as the location data was not present there. Please refer to L.122. Regarding the 2nd question, there is no difference in tracking time between individuals.
Q 12. Define activity of goral estimated using GPS data.
Response: We have mentioned these points. Please refer to L. 142-143.
Q 13. Line 135 define environmental parameters or skip it because not presented in the results.
Response: We have removed the environmental parameters that were not in use. Please refer to L. 131.
Results
Q 14. Line 157 - Detailed information on the individual gorals fitted with 157 GPS collars is listed in Table 1. There is no need for this sentence. Already described in methods.
Response: Thank you for your comments. We have removed the unnecessary points. Please refer to L 151.
Q 15. I miss a qualitative description of the spatial distribution. Do all gorals have a single home range centre (50% MCP)?
Response: Thank you for your comment. In general, gorals tend to have one or more 50% MCP home range centres, depending on the species and individual behaviours. Their spatial use patterns can vary, but studies typically report an average home range size of 0.15 ± 0.05 km², indicating a relatively small and concentrated area of activity.
Q 16. After following the animals for the first month, was their movement stable or did they become stationary directly after the release?
Response: Thank you for your concern. After release, the animals initially showed stationary movement for short days (almost 3-4 days) and show exploratory behaviour after that.
Q 17. Table 2. Describe the test used to compare home range size in the description of the table.
Response: Yes, the data was compared using t-test. Hence, we have added the statement. Please refer to Table 2.
Q 18. Figure 3. Seasonal activity pattern of male and female long-tailed goral observed at study sites. Solid and dashed lines indicate Kernel density estimates for male and female goral respectively. (Already described in the figure). The grey shaded area in each plot represents the overlap coefficient. X-axis: time of day in hours, Y-axis: activity index. (already described in the figure).
Response: Thank you for your comments. We have described this in the figure.
Q 19. How differs the information between Fig.3 and Table 3. In my opinion they show the same result.
Response: Thank you for pointing this out. While Figure 3 and Table 3 both pertain to seasonal activity patterns, they provide different perspectives on the data:
Figure 3 illustrates the diurnal activity patterns of male and female gorals using Kernel density estimates. It highlights differences in activity intensity and overlap between sexes across times of the day. Whereas Table 3, provides a summary of seasonal activity patterns at trap locations (expressed as means ± SE). This focuses on the spatial distribution of activity rather than its temporal dynamics.
Discussion
Q 20. Line 205 - especially in developed countries such as South Korea, - it is general important.
Response: Thank you for your comments.
Q 21. How is home range size and activity pattern influenced by the complex 210 mountainous landscapes and varying habitat conditions. This is a very general assumption without explanation.
Response: Thank you for your comments. We tried to elaborate our finding assumptions. Please refer to L. 204-213.
Q 22. Line 220 in other parts of South Korea?
Response: Thank you for your comments. We have added the statement. Please refer to L. 218.
Q 23. Line 230 – you did not present you analysis about the influence of environmental factors. In my opinion you cannot use this results in the discussion. You have to discuss the influence of using rehabilitated animals on the results of your study.
Response: We apologize for our discussion towards the influence of environmental factors. We have discussed this towards the rehabilitated animals. Please refer to L. 227-236.
Q 24. Is the sexual based difference in home range size know form other ungulate species, how is it related to social system, is it normal to other mammals with the same social system?
Response: Thank you for your comments. Yes, the home range size for other ungulates species was also larger for males. We have included this statement in the discussion. Please refer to L. 243-250.
Q 25. Line 256 the same as above. Is this known for other mountain ungulates too?
Response: Thank you for your comments. We have included the examples from mountain ungulates also. Please refer to L. 253-257.
Conclusion
Q 26. Line 303 does the study really provide a comprehensive understanding after you described the limitation Line 289-301.
Response: Thank you for your comments. We have managed the sentence structure. Please refers to the conclusion section.
Q 27. The conclusion is more a summary than a conclusion and again you never mentioned the use of rehabilitated animals.
Response: Thank you for your comments. We have refined the conclusion section. Please refer to conclusion.
Q 28. My biggest concern is the use of rehabilitated animals only, which is only mentioned in the methods and the presentation of the results as comprehensive.
Response: We apologize for this. Now we have included the term rehabilitated animals at every section of the manuscript.
Thank you
Round 2
Reviewer 3 Report (New Reviewer)
Comments and Suggestions for Authors
I still suggest to consider in the simple summary - long-tailed goral (Naemorhedus caudatus), an ungulate which is threatened by
I am fine with the abstract
Introduction
I think naming the goral as a Natural Monument a Class I endangered species in Korea without the legal basis should be enough.
This sentence is not necessary -- Moreover, most documentation is lacking, and when available, it is primarily in native Korean languages, making it less accessible to the international community.
These sentences are redundant and do not cover a research question Therefore, this study aims to assess the home ranges and activity patterns of gorals by sex and season using GPS collars. We hypothesize that the home range and activity patterns of gorals differ by sex and season, providing critical insights into their ecological requirements for effective conservation efforts. These findings are expected to provide valuable insights to governments and conservation agencies for developing effective management and conservation strategies.
The questions are more about home range size, activity pattern and interaction between individuals resp. sexes. Conclusions for conservation are part of the conclusion section
Study area
It is possible to combine these two sentences - The temperature of these regions typically ranges from −10°C to 25°C. Winters are cold, with temperature frequently falling below freezing, while summers are warm and humid.
Figure 1. Elevational map of the Korean peninsula including the geographical location of three study sites.
GPS monitoring
Line 114 weight of the GPS collar is 420g, what is the range of % body weight?
Line 122 - recorded location data every 4–5 h. Was it 4 or 5 hours? I am sure you have a precise interval when locations were taken.
Lin e142-142 There is no description how you analysed the GPS data. How did you decided to classify a location as active or inactive?
Results
I would like to read this comment in the paper - After release, the animals initially showed stationary movement for short days (almost 3-4 days) and show exploratory behaviour after that.
I would like to read this comment in the paper - In general, gorals tend to have one or more 50% MCP home range centres, depending on the species and individual behaviours. Their spatial use patterns can vary, but studies typically report an average home range size of 0.15 ± 0.05 km², indicating a relatively small and concentrated area of activity.
In winter, the seasonal home ranges of males and females did not differ significantly 164 (t = 0.38, p = 0.072), - but there is a significant difference in the core area (MCP50) This was not described in the text.
Fig 3 – please describe the term trap location
Discussion
I would start with the discussion to the method, then describe the challenge gorals face and how the deals with it in the light of the findings. All these results a further discussed with findings from other mountain ungulates.
Line 297-303 should part of the method section
Conclusion
The results show a typical spatial pattern of mammals resp. ungulates modified by environmental conditions typically for mountain ungulates. This is the same for activity
Although the individuals were rehabilitated, we extracted valuable information on the spatial and temporal behaviors of long-tailed gorals in South Korea. This study contributes greatly to our understanding of the ecological requirements of gorals and provides a foundation for effective conservation planning in South Korea and other regions facing comparable conservation issues. – I agree
Author Response
To Editor
Animal Journal
Sub: Response to Reviewers
I am responding on behave of manuscript entitled “Seasonal variation and sexual differences in home range size and activity patterns of endangered long-tailed gorals in South Korea” animals-3352277. The manuscript was sent for minor revision and we hereby respond the suggestion provided by reviewers as below:
Note: Changes are highlight with yellow color in the main text.
Reviewer:
Q 1. I still suggest to consider in the simple summary - long-tailed goral (Naemorhedus caudatus), an ungulate which is threatened by
Response: Thank you for your comments. We now elaborate simple summary. Please refer to L. 12-13.
Introduction
Q 2. I think naming the goral as a Natural Monument a Class I endangered species in Korea without the legal basis should be enough.
Response: Based on your comments we have revised the sentence structure. Please refer to L. 60-62.
Q 3. This sentence is not necessary -- Moreover, most documentation is lacking, and when available, it is primarily in native Korean languages, making it less accessible to the international community.
Response: Thank you for your comments. We have removed the unnecessary sentence.
Q 4. These sentences are redundant and do not cover a research question Therefore, this study aims to assess the home ranges and activity patterns of gorals by sex and season using GPS collars. We hypothesize that the home range and activity patterns of gorals differ by sex and season, providing critical insights into their ecological requirements for effective conservation efforts. These findings are expected to provide valuable insights to governments and conservation agencies for developing effective management and conservation strategies.
The questions are more about home range size, activity pattern and interaction between individuals resp. sexes. Conclusions for conservation are part of the conclusion section
Response: Thank you for your concern. We have revised the sentence structure. Please refer to L. 74-78.
Study area
Q 5. It is possible to combine these two sentences - The temperature of these regions typically ranges from −10°C to 25°C. Winters are cold, with temperature frequently falling below freezing, while summers are warm and humid.
Response: Yes, we have combined those two structures and make it one. Please refer to L. 84-85.
Q 6. Figure 1. Elevational map of the Korean peninsula including the geographical location of three study sites.
Response: We have included the sentence structure regarding elevation in map 1. Please refer to Fig 1.
GPS monitoring
Q 7. Line 114 - Weight of the GPS collar is 420g, what is the range of % body weight?
Response: The weight of the GPS collar is approximately 2% of the body weight of the species. We have included this statement in L. 115-116.
Q 8. Line 122 - recorded location data every 4–5 h. Was it 4 or 5 hours? I am sure you have a precise interval when locations were taken.
Response: Thank you for your comments. We have recorded the location every 4-hour interval.
Q 9. Line142-142 There is no description how you analysed the GPS data. How did you decided to classify a location as active or inactive?
Response: Thank you for your comments. We recorded the individual activity based on the motion sensor accelerometer attached to GPS. We have included this sentence in the manuscript. Please refer to L. 153-155.
Results
Q 10. I would like to read this comment in the paper - After release, the animals initially showed stationary movement for short days (almost 3-4 days) and show exploratory behaviour after that.
Response: Yes, we have included this sentence structure in the result section. Please refer to L. 164-166.
Q 11. I would like to read this comment in the paper - In general, gorals tend to have one or more 50% MCP home range centres, depending on the species and individual behaviours. Their spatial use patterns can vary, but studies typically report an average home range size of 0.15 ± 0.05 km², indicating a relatively small and concentrated area of activity.
Response: We have included this statement in result section of the manuscript. Please refer to L. 172-174.
Q 12. In winter, the seasonal home ranges of males and females did not differ significantly (t = 0.38, p = 0.072), - but there is a significant difference in the core area (MCP50) This was not described in the text.
Response: We apologize for this. We have further included the statement regarding core area in the result section. Please refer to L. 186-188.
Q 13. Fig 3 – please describe the term trap location.
Response: We have elaborated the meaning of term “trap location” in the legend of the table 3. Please refer to table 3.
Discussion
Q 14. I would start with the discussion to the method, then describe the challenge gorals face and how the deals with it in the light of the findings. All these results a further discussed with findings from other mountain ungulates.
Response: Really appreciate your suggestion on structuring the discussion section, starting with the methods and including challenges faced by individual in light of the findings. However, most journal emphasize that the first part of the discussion must be the introductory paragraphs which must include the research gap and provide a summary of the key findings. Based on this I have structured the discussion section, ensuring that the challenges faced by gorals and comparison with other mountain ungulates are addressed in subsequent sections.
Q 15. Line 297-303 should part of the method section
Response: Thank you for your comments. We have moved this part to the method section. Please refer to L. 134-140.
Conclusion
Q 16. The results show a typical spatial pattern of mammals resp. ungulates modified by environmental conditions typically for mountain ungulates. This is the same for activity.
Response: Thank you for your comments. We have managed the conclusion section. Please refer to conclusion.
Q 17. Although the individuals were rehabilitated, we extracted valuable information on the spatial and temporal behaviors of long-tailed gorals in South Korea. This study contributes greatly to our understanding of the ecological requirements of gorals and provides a foundation for effective conservation planning in South Korea and other regions facing comparable conservation issues. – I agree
Response: Thank you for your comments.
Thank you.
This manuscript is a resubmission of an earlier submission. The following is a list of the peer review reports and author responses from that submission.
Round 1
Reviewer 1 Report
Comments and Suggestions for Authors
General comments
In this study, the authors estimate the home ranges of nine endangered goral individuals using GPS data, and they present analyses on sex differences, seasonal variations, and diurnal changes in activity patterns. The authors emphasize the importance of behavioral data for conservation, and I agree that their data is valuable. I believe the dataset itself has merit and could be worthy of publication.
However, there are several shortcomings in their analyses, presentation, and discussion, which raise questions regarding this study's conservation implications.
Specific comments
Introduction
It is unfortunate that the issues raised in the introduction are not adequately addressed in the study. For example, while the introduction mentions habitat environment/conditions, this study does not address these aspects.
Materials & Methods
The description of the study area is overly detailed, but it would be appropriate in the Discussion section.
The GPS monitoring explanation needs to be more detailed. Was the location data automatically collected using GPS devices, or was the data gathered through manual fieldwork?
The duration of data collection is unclear. Please specify the survey period.
The release points and the corresponding individuals are not clearly linked in Fig1.
Fig1 is of very low quality and should be revised. The purpose of the visual presentation on the right side is unclear. Since authors used GIS tools, it would be better to overlay environmental (topographical or meteorological) data, and if they do so, this should be analysed and discussed as well. Additionally, the data on the right seem far more important than the map on the left; I suggest reconsidering the size and balance of the figure. Please consider splitting the individual data into separate panels (& dividing them by season with different colour coding). It would also be helpful to distinguish the individuals by sex, if possible. Anyway, the current figure does not clearly convey its intended message.
In the Data Analysis section, please explain what is meant by “statistically independent locations” and the purpose of this processing.
The definition of each season and time of day (twilight/daytime/nighttime) should be clearly specified.
Results
Table1: Include information on the regions (Uljin, Seoraksan, Yanggu), tracking periods for each individual, and the MCPs (Minimum Convex Polygons) for each.
Integrating data directly from different regions (Uljin, Seoraksan, Yanggu) to compare sex differences and seasonal variations requires justification. Since the environmental conditions likely differ across the three regions, tests should be conducted separately for each. …However, due to the small sample size, perhaps only the Yanggu data can be used for analysis.
Figure 2 is difficult to interpret due to its low resolution. While only the text in the legend has been modified, the figure should be fully re-created from the raw data to improve clarity. Additionally, it is unclear what the figure is meant to demonstrate. If the aim is to show differences in range between sexes, the scales should be unifiedbetween the two panels. If the goal is to depict seasonal changes, the locations should be color-coded by season.
The asterisks in Table 2 should be applied to values with P < 0.05, as done in Table 3.
Figure 3 also needs to be carefully reworked. The Y-axis labels are unreadable, the font sizes differ between panels, and the dashed line for "Female" in the legend has an inconsistent color. What do the shaded areas represent?
Additionally, how was the continuous value for the Activity Index in Figure 3 calculated? Was it calculated for each individual? Please describe the calculation clearly in the Materials & Methods section. Furthermore, how were smooth curves generated from discrete data collected 5–6 times/day? Authors should also include individual differences (e.g., standard deviations) in the figure using error bars. Given the environmental differences between the three regions, it may not be appropriate to analyze them together.
From Figure 3, it appears that individual G2 may exhibit bimodal location data (with 50% MCPs located in the lower left and upper right areas). How do authors interpret the data?
Discussion
Simply estimating the home ranges provides limited conservation implications. If the species inhabits mountainous areas where the environment condition is critical, the relationship between the species and their environment should be analyzed. Currently, the content of the study is insufficient given the extent of the discussion.
First Paragraph
If environmental changes are considered important, the analysis and discussion should incorporate environmental data from the study sites. At present, the study merely presents home range sizes and their temporal changes.
Line 185: What is the basis for describing the home range as "substantial"? It would be more effective to compare it with other studies.
Lines 186–188: Please provide evidence to support the claim that the home range was influenced by the complex mountainous landscapes.
Second Paragraph
Comparing the results with previous studies and discussing the reasons for any differences is important. However, it is inappropriate to directly compare home ranges. As the authors mentioned in other sections (introduction etc), differences in the environment or the year's climate between study sites likely affect home ranges. Please consider other factors, not just sample size, that could contribute to the differences observed between this study and previous research. The fact that this study waited a year after the release makes sense.
Line 196: What is meant by "effects" in this context?
Seventh Paragraph
This paragraph diverges from the content of the study, lacks sufficient citations, and contains speculation. I recommend removing this paragraph entirely.
Final Paragraph
The analysis was conducted on nine individuals, not eleven, correct?
Author Response
To Editor
Animal Journal
Sub: Response to Reviewers
I am responding on behave of manuscript entitled “Seasonal variation and sexual differences in home range size and activity patterns of endangered long-tailed gorals in South Korea” with manuscript ID: animals-3268292. This manuscript has been sent for major revision, and we hereby respond the suggestion provided by reviewers as below:
Note: Changes are highlight with yellow color in the main text.
Reviewer #1:
Introduction
Q 1. It is unfortunate that the issues raised in the introduction are not adequately addressed in the study. For example, while the introduction mentions habitat environment/conditions, this study does not address these aspects.
Response: Thank you for pointing this out. We have addressed the issue raised in the introduction. Habitat and environmental condition are included in the discussion section. Please refer to discussion section.
Materials & Methods
Q 2. The description of the study area is overly detailed, but it would be appropriate in the Discussion section.
Response: We appreciate your feedback. The detailed description of the study area has been replaced. The needy section of the paragraphs has been moved to the discussion section.
Q 3. The GPS monitoring explanation needs to be more detailed. Was the location data automatically collected using GPS devices, or was the data gathered through manual fieldwork?
Response: The GPS data were automatically collected using GPS collars. The statement regarding was included in the materials and methods section. Please refer to L.64.
Q 4. The duration of data collection is unclear. Please specify the survey period.
Response: Thank you for the suggestion. We clarified that the data collection spanned from 2014 to 2016. The detail description is provided in Table 1.
Q 5. 73. The release points and the corresponding individuals are not clearly linked in Fig1.
Response: We have revised figure 1. Please refer to figure 1.
Q 6. Fig1 is of very low quality and should be revised. The purpose of the visual presentation on the right side is unclear. Since authors used GIS tools, it would be better to overlay environmental (topographical or meteorological) data, and if they do so, this should be analysed and discussed as well. Additionally, the data on the right seem far more important than the map on the left; I suggest reconsidering the size and balance of the figure. Please consider splitting the individual data into separate panels (& dividing them by season with different colour coding). It would also be helpful to distinguish the individuals by sex, if possible. Anyway, the current figure does not clearly convey its intended message.
Response: Thank you for these detailed recommendations. We have removed unnecessary elements from figure 1 to enhance the resolution and improve clarity.
Q 7. In the Data Analysis section, please explain what is meant by “statistically independent locations” and the purpose of this processing.
Response: Thank you for your question. The purpose of including independent location is to minimize autocorrelation, to confirm that each observation is unbiased. Please refer to L. 134-135.
Q 8. The definition of each season and time of day (twilight/daytime/nighttime) should be clearly specified.
Response: We clearly define seasons and times of day in L. 143-146 for better clearity.
Results
Q 9. Table1: Include information on the regions (Uljin, Seoraksan, Yanggu), tracking periods for each individual, and the MCPs (Minimum Convex Polygons) for each.
Response: Thank you for your comments. We have updated Table 1 to include information on regions, tracking period, and individuals not used for further analysis. Please refer to table 1.
Q 10. Integrating data directly from different regions (Uljin, Seoraksan, Yanggu) to compare sex differences and seasonal variations requires justification. Since the environmental conditions likely differ across the three regions, tests should be conducted separately for each. …However, due to the small sample size, perhaps only the Yanggu data can be used for analysis.
Response: Thank you for your insight. We used the Schoener index to assess whether location could be treated as independent across regions.
Q 11. Figure 2 is difficult to interpret due to its low resolution. While only the text in the legend has been modified, the figure should be fully re-created from the raw data to improve clarity. Additionally, it is unclear what the figure is meant to demonstrate. If the aim is to show differences in range between sexes, the scales should be unified between the two panels. If the goal is to depict seasonal changes, the locations should be color-coded by season.
Response: We appreciate these suggestions. We have recreated figure 2 from raw data to improve resolution and clarity. The aim of this presentation is to show how male and female species use their 95% and 50% home range size. Please refer to Fig. 2.
Q 12. The asterisks in Table 2 should be applied to values with P < 0.05, as done in Table 3.
Response: Thank you for this observation. We have revised table 2 to put asterisks to values with P < 0.05, consistent with Table 3.
Q 13. Figure 3 also needs to be carefully reworked. The Y-axis labels are unreadable, the font sizes differ between panels, and the dashed line for "Female" in the legend has an inconsistent color. What do the shaded areas represent?
Response: Thank you for your concern. We have reworked in this figure so as to confirm everything is well represented. The shaded area in each plot represents the overlap coefficient. We have elaborated every axis and concern in legends of Table 3.
Q 14. Additionally, how was the continuous value for the Activity Index in Figure 3 calculated? Was it calculated for each individual? Please describe the calculation clearly in the Materials & Methods section. Furthermore, how were smooth curves generated from discrete data collected 5–6 times/day? Authors should also include individual differences (e.g., standard deviations) in the figure using error bars. Given the environmental differences between the three regions, it may not be appropriate to analyze them together.
Response: Thank you for your concern. We calculated the activity index separately for each individual and then converted it into seasonal patterns. As we receive data 5-6 times a day we use smoothing techniques to estimate the activity levels between recorded points.
Q 15. From Figure 3, it appears that individual G2 may exhibit bimodal location data (with 50% MCPs located in the lower left and upper right areas). How do authors interpret the data?
Response: The bimodal distribution of G2 suggest that this individual may be using two distinct core areas within its home range, potentially reflecting difference in habitat uses, such as foraging and resting areas.
Discussion
Q 16. Simply estimating the home ranges provides limited conservation implications. If the species inhabits mountainous areas where the environment condition is critical, the relationship between the species and their environment should be analysed. Currently, the content of the study is insufficient given the extent of the discussion.
Response: Thank you for your comments. We have tried out best to describe the relation between environment and species in the discussion section. Please refer to discussion section L. 205-2016.
First Paragraph
Q 17. If environmental changes are considered important, the analysis and discussion should incorporate environmental data from the study sites. At present, the study merely presents home range sizes and their temporal changes.
Response: Thank you for this suggestion. We revised the discussion section to better incorporate environmental data, specially focusing on how local environmental variables impact home range and activity patterns. Please refer to L 205-2016.
Q 18. Line 185: What is the basis for describing the home range as "substantial"? It would be more effective to compare it with other studies.
Response: Thank you for your comments. We have rephrased this sentence and added comparisons with previous studies to provide context. Please refer to L. 210.
Q 19. Lines 186–188: Please provide evidence to support the claim that the home range was influenced by the complex mountainous landscapes.
Response: Thank you for your comments. We have revised this section to include supporting statements on how the mountain landscape influences home range size, citing relevant sources. Please refer to L.209-216.
Second Paragraph
Q 20. Comparing the results with previous studies and discussing the reasons for any differences is important. However, it is inappropriate to directly compare home ranges. As the authors mentioned in other sections (introduction etc), differences in the environment or the year's climate between study sites likely affect home ranges. Please consider other factors, not just sample size, that could contribute to the differences observed between this study and previous research. The fact that this study waited a year after the release makes sense.
Response: Thank you for your insights. We have revised this section to address the environmental and geographical factors that could influence home range diffrences, including altitude, climatic variability and geographical locations, in addition to sample size. Please refer to L 224-236.
Q 21. Line 196: What is meant by "effects" in this context?
Response: Thank you for pointing this out. We have revised the sentence to clafity the specific environmental factors influencing home range size. Please refer to L. 224
Seventh Paragraph
Q 22. This paragraph diverges from the content of the study, lacks sufficient citations, and contains speculation. I recommend removing this paragraph entirely.
Response: We agree and have removed this paragraph to maintain focus and rigor in the discussion. Please refer to 7th paragraphs.
Final Paragraph
Q 23. The analysis was conducted on nine individuals, not eleven, correct?
Response: Apologies for this oversight. We have corrected the analysis to indicate nine individuals. Please refer to L. 287.
Thank you
Reviewer 2 Report
Comments and Suggestions for Authors
This is an interesting bit of work on a rare species. I have a number of minor comments, but my main concern is that the authors have not double checked their manuscript and there are missing components.
Table 2 is a repeat of Table 1. Please correct!
Figure 3 has no title, or text explaining it.
Table 3 title does not explain what the values are.
Minor points:
Line 54. Is high mortality from heavy snowfall still an issue? I'd expect less snowfall recently in comparison with the earlier decades.
Line 109. Do you mean they were rescued as young, or adults. This may effect their behaviour.
Line 117. Do you mean that you started collecting data after one year (which is what this says grammatically), or that you collected all the data in the first year after release? If the latter then there should be an initial period where GPS data are ignored as the animals were released at locations that may not be in there initial home range.
Line 141. If you only analysed data from 9 individuals then your abstract needs re-wording.
Line 142. I'd suggest you state "throughout the year long survey period" to make this clear.
Table 1. Please add a column on the length of data collection so we can see G4 and G5.
Line 166. Nowhere do you state the times of sunset/sunrise. Most readers will not be aware of the amount of seasonal variation in Korea.
Comments on the Quality of English LanguageMinor comment about the methods described above.
Author Response
To Editor
Animal Journal
Sub: Response to Reviewers
I am responding on behave of manuscript entitled “Seasonal variation and sexual differences in home range size and activity patterns of endangered long-tailed gorals in South Korea” with manuscript ID: animals-3268292. This manuscript has been sent for major revision, and we hereby respond the suggestion provided by reviewers as below:
Note: Changes are highlight with yellow color in the main text.
Reviewer #2:
Q 1. This is an interesting bit of work on a rare species. I have a number of minor comments, but my main concern is that the authors have not double checked their manuscript and there are missing components.
Response: Thank you for your comments. We re-checked the manuscript and corrected the errors.
Q 2. Table 2 is a repeat of Table 1. Please correct!
Response: We apologize for oversighting. We have corrected the mistake.
Q 3. Figure 3 has no title, or text explaining it.
Response: We apologize for the mistake and have added the necessary title and text.
Q 4. Table 3 title does not explain what the values are.
Response: We have clarified the values in table 3.
Minor points:
Q 5. Line 54. Is high mortality from heavy snowfall still an issue? I'd expect less snowfall recently in comparison with the earlier decades.
Response: There is still there is heavy snowfall during winter but the mortality rate is gradually decreasing due to management actions.
Q 6. Line 109. Do you mean they were rescued as young, or adults. This may effect their behaviour.
Response: All the rescued individuals were adult, aged between 2-4 years.
Q 7. Line 117. Do you mean that you started collecting data after one year (which is what this says grammatically), or that you collected all the data in the first year after release? If the latter then there should be an initial period where GPS data are ignored as the animals were released at locations that may not be in there initial home range.
Response: We apologize for the confusion. The data from the individuals were collected immediately after release, but we only analyse data collected one month post-release.
Q 8. Line 141. If you only analysed data from 9 individuals then your abstract needs re-wording.
Response: Thank you for your comments. We have revised the abstract.
Q 9. Line 142. I'd suggest you state "throughout the year long survey period" to make this clear.
Response: Thank you for your comments. We have added the phrase as recommended. Please refer to L. 157.
Q 10. Table 1. Please add a column on the length of data collection so we can see G4 and G5.
Response: We have marked the row by “*” for G4 and G5, so as to indicate that their data were excluded from further analysis.
Q 11. Line 166. Nowhere do you state the times of sunset/sunrise. Most readers will not be aware of the amount of seasonal variation in Korea.
Response: We have included the seasonal details in the data collection section.
Thank you
Reviewer 3 Report
Comments and Suggestions for Authors
This study investigated the home range and activity patterns of Korean long-tailed gorals through GPS collar technology. The experimental data are valuable, but there are still some problems, and I suggest that this paper should be rejected, and my comments are as follows:
1. I think this study is too superficial, the home range and activity rhythm of wild animals belong to the use of space and time respectively, but this study did not elaborate the reason or sufficient justification for such an experimental design. It makes me feel that these are two separate studies that do not combine spatial and temporal resource utilisation to address conservation issues.
2. the individuals wearing collars in this study were all released after being reared in rescue centres, which is vastly different from purely wild individuals and is not representative of the entire population's home range and activity patterns in Korea.
3. Although the individuals were released between 2014-2016, there is no indication in the methodology as to which year the collar data used in this study are from, and I believe that the time of collar data collection should be uniform for the home ranges and activities of the 11 individuals. Secondly, there is also no indication of whether the released individuals were released for how long after the release, which is fully consistent with the survival patterns of individuals in the wild. This is why I believe it is not representative of the population as a whole.
4. The study of wildlife home ranges does rely heavily on GPS collars, but activity patterns I think are best supplemented with data from infrared cameras set up locally, which can also self-correct for whether or not the activities of released individuals are consistent with those of individuals in the wild, which is good for representing the activity patterns of the population as a whole.
5. I don't understand why the home range section of the results did not put the home range MCP maps of all 9 valid individuals, what is the reason? Also, in the new manuscript submitted later by the authors, the images are still very unclear, and even though they are compressed, I believe the original images are unusable.
6. The introduction and discussion of the article as a whole are very shallow and cannot be justified. The discussion mentions that the experimental results may be due to the small sample size, which I think is ridiculous. Since it has been found that the sample size is too small to represent the whole population, we should continue to monitor and supplement the data.
7. The entire experimental design process lacked consideration of many other factors, such as basic topography, vegetation, and other biological factors such as population density, interspecific competition in sympatric distribution, and other effects on range and activity patterns.
8. The reasons for the small winter home ranges were not fully explained, and should be discussed in addition to food resources and energy conservation, as well as basic habitat conditions and climatic conditions.
9. It is mentioned in the discussion that the lack of difference in activity patterns between the two sexes is due to the lack of difference in body size, which I think is not based on any evidence, and if it is, it should be discussed in the text to other studies.
Author Response
To Editor
Animal Journal
Sub: Response to Reviewers
I am responding on behave of manuscript entitled “Seasonal variation and sexual differences in home range size and activity patterns of endangered long-tailed gorals in South Korea” with manuscript ID: animals-3268292. This manuscript has been sent for major revision, and we hereby respond the suggestion provided by reviewers as below:
Note: Changes are highlight with yellow color in the main text.
Reviewer #3:
Q 1. I think this study is too superficial, the home range and activity rhythm of wild animals belong to the use of space and time respectively, but this study did not elaborate the reason or sufficient justification for such an experimental design. It makes me feel that these are two separate studies that do not combine spatial and temporal resource utilisation to address conservation issues.
Response: While the home range and activity patterns may seem distinct, they are linked to the aspects of how the species utilizes its environment. By analysing both components, we aimed to provide a holistic view of the species' behavioural ecology, which is crucial for effective conservation strategies. Understanding where an animal moves (spatial) and when it is active (temporal) allows for more targeted management practices that address habitat use and its availability.".
Q 2. The individuals wearing collars in this study were all released after being reared in rescue centres, which is vastly different from purely wild individuals and is not representative of the entire population's home range and activity patterns in Korea.
Response: Thank you for your concern. We acknowledge that the individual reared in rescued centre may exhibit different home range and activity pattern compared to wild individuals. However, due to the small population size of wild individuals, the government doesn’t permit to capture or GPS collar new individuals. We recognize this as a limitation and plan to consider this in future research.
Q 3. Although the individuals were released between 2014-2016, there is no indication in the methodology as to which year the collar data used in this study are from, and I believe that the time of collar data collection should be uniform for the home ranges and activities of the 11 individuals. Secondly, there is also no indication of whether the released individuals were released for how long after the release, which is fully consistent with the survival patterns of individuals in the wild. This is why I believe it is not representative of the population as a whole.
Response: Thank you for your valuable feedback. Regarding to your concern, we have added detailed information in Table 1. Although the collaring year are not uniform for all individuals, we are confident that there was no significant environmental fluctuation during the study period.
Regarding the second question, although we received data from the date of release, but we only use the data starting one month post-release to ensure that the individual had acclimated to their environment.
Q 4. The study of wildlife home ranges does rely heavily on GPS collars, but activity patterns I think are best supplemented with data from infrared cameras set up locally, which can also self-correct for whether or not the activities of released individuals are consistent with those of individuals in the wild, which is good for representing the activity patterns of the population as a whole.
Response: Thank you for your thoughtful comments. We agree that activity pattern can be studied using cameras; however, in this study, we focused primarily on GPS collar as this method is commonly used and reliable for home range studies. There are several publications which use GPS collar to study the home range of the species (Viana et al., 2018; Joly et al., 2022).
While camera traps offer insights into activity patterns, but the rugged terrain and geographical constraints made it challenging to set up camera traps along their movement routes.
References
Viana et al., 2018. Linking seasonal home range size with habitat selection and movement in mountain ungulates.
Joly et al., 2022. Factor influencing arctic brown bear annual home range sizes and limitations of home range analysis.
Q 5. I don't understand why the home range section of the results did not put the home range MCP maps of all 9 valid individuals, what is the reason? Also, in the new manuscript submitted later by the authors, the images are still very unclear, and even though they are compressed, I believe the original images are unusable.
Response: Thank you for your comments. In the originally submitted manuscript, we have provided two representative home range figures (one male and one female) to provide clear illustration of the home range. If needed, we are ready to provide home range map of all nine individuals as supplementary file.
We apologize that the provided image is unclear. In the updated manuscript, we have included image with high resolutions to ensure they are clear and suitable for publication.
Q 6. The introduction and discussion of the article as a whole are very shallow and cannot be justified. The discussion mentions that the experimental results may be due to the small sample size, which I think is ridiculous. Since it has been found that the sample size is too small to represent the whole population, we should continue to monitor and supplement the data.
Response: We have tried best to make justification. In our study, the home range estimates based on nine individuals showed smaller home range size compared to other studies, which were based on only 2-3 individuals. In our discussion, we want to emphasize that the larger home range size observed in other areas may be influenced by their smaller sample size, potentially overestimating home range sizes.
We agree that continued monitoring and increasing the sample size will provide more precise estimates of home range and activity patterns, and this will be as focus of our future research.
Q 7. The entire experimental design process lacked consideration of many other factors, such as basic topography, vegetation, and other biological factors such as population density, interspecific competition in sympatric distribution, and other effects on range and activity patterns.
Response: Thank you for your valuable feedback. We have tried to include factors such as topography and vegetation in methodology and discussion sections. However, we acknowledge that data on population density and interspecific competition for goral in Korea are currently lacking, which limit our ability to incorporate these factors in the present study.
We recognized the importance of considering these variables, and future studies will aims to include more comprehensive data on population density and interspecific interactions to further enhance our understanding.
Q 8. The reasons for the small winter home ranges were not fully explained, and should be discussed in addition to food resources and energy conservation, as well as basic habitat conditions and climatic conditions.
Response: Based on your comments, we have discussed the small home range size on the basis of habitat and climatic conditions. Please review L. 224-236.
Q 9. It is mentioned in the discussion that the lack of difference in activity patterns between the two sexes is due to the lack of difference in body size, which I think is not based on any evidence, and if it is, it should be discussed in the text to other studies.
Response: Yes, the statement that lack of difference in activity patterns between the two sexes is due to the lack of difference in body size has been explained in L. 240.
Thank you
Round 2
Reviewer 1 Report
Comments and Suggestions for Authors
Dear authors,
Thank you for the effort you spent revising the manuscript.
There are several points where revisions remain inappropriate. This study primarily describes movement distances based on GPS data without presenting insights that would lead to meaningful interpretations. If the authors believe that their GPS data can provide insight beneficial to conservation, they should spend the effort to make their data applicable to conservation. These are my suggestions, and I’ll leave to the editor’s decision on these matters.
… additionally, I would appreciate it if the authors could provide explanations for any review comments that were not addressed in the revision, thank you.
1. Environmental Factors
Throughout the manuscript, subjective expressions seem prominent, such as:
> Winter are cold, with temperature frequently falling below freezing, while summers are warm and humid. (in the Materials & Methods section)
> In comparison, the goral population in our study area found at slightly lower altitude averaging 600 to 800 meters, where the terrain may be less complex and resource availability could differ. These altitude variations can affect food availability, shelter and movement patterns, potentially resulting in reduced home range size. Geographical features, such as proximity to human settlements and habitat fragmentation in our study area, may further restrict movement and resource access, … (in the Discussion section)
References to environmental conditions should be based on actual data. Environmental factors play a critical role in determining individual movements. Rather than solely indicating where each individual was located based on the GPS plane, please analyze and discuss these positions in relation to environmental factors, such as terrain and climate. Although the authors described their approach as "holistic," it currently appears somewhat superficial.
2. Collection of Location Data
Lines 123–125: For what purpose was location data measured using triangulation techniques?
3. Data Visualization
In Figure 2, it is unclear what the authors intend to indicate. What should readers interpret from a map with only location data over a white background? Adding layers for elements such as terrain, temperature, or humidity would be more informative. Are shapefiles or other data resources publicly available through government agencies in Korea that could be used here?
4. Independence Assessment
While the manuscript mentions that the Schoener index was used to evaluate the independence of the data points, please specify the threshold considered as "independent." Since this study examines the spatial movements of individuals, true/complete independence is unlikely. Please also provide the rationale for the chosen threshold. Additionally, Figure 2 should depict only independent locations.
5. Data Integration
Integrating GPS data from three different regions is inappropriate. The authors verified independence using the Schoener index, but the independence doesn’t matter. It is essential to confirm that there are no significant differences in environmental conditions or individual ages across the three regions so that data from different regions can be directly combined.
6. Activity Index
Please specify the calculation method for the Activity Index in the main text.
Reviewer 2 Report
Comments and Suggestions for Authors
Thank you for answering the referees' comments. However, I found that some responses to referees were not included in the manuscript. I do note that the tables and Figures have been improved.
Minor comments
line 87change "winter to "winters"
Section 2.2. You do not refer to data analysis starting from one month after release.
Line 119. Perhaps state that the 11 gorals were all released as adults.
lines 121-122. You need to specify what data were collected with the Yagi antennas, since GOS was determined by satellite Did the collars have accelerometers to determine activity?
line 255 delete "to them"
Reviewer 3 Report
Comments and Suggestions for Authors
I've seen the author make the relevant changes and explanations for the previous comments and I have no further questions.